# A meta-analysis of pre-pregnancy maternal body mass index and placental DNA methylation identifies 27 CpG sites with implications for mother-child health

Nora Fernandez-Jimenez [1,40], Ruby Fore[2,40], Ariadna Cilleros-Portet[1,40], Johanna Lepeule [3], Patrice Perron[4], Tuomas Kvist[5], Fu-Ying Tian[6], Corina Lesseur [7], Alexandra M. Binder[8,9], Manuel Lozano[10,11], Jordi Martorell-Marugán [12,13], Yuk J. Loke[14,15], Kelly M. Bakulski [16], Yihui Zhu [17], Anne Forhan[18], Sara Sammallahti[19], Todd M. Everson [6,20], Jia Chen[7], Karin B. Michels[9,21], Thalia Belmonte [22], Pedro Carmona-Sáez [12,13], Jane Halliday[14,15], M. Daniele Fallin[23], Janine M. LaSalle [17], Jorg Tost [24], Darina Czamara [25], Mariana F. Fernández [26,27,28], Antonio Gómez-Martín [27,29], Jeffrey M. Craig[14,30], Beatriz Gonzalez-Alzaga[27,29], Rebecca J. Schmidt [31], John F. Dou[16], Evelyne Muggli [14,15], Marina Lacasaña [29,28,27], Martine Vrijheid [28,32,33], Carmen J. Marsit [6,20], Margaret R. Karagas [34], Katri Räikkönen[5], Luigi Bouchard[34,35], Barbara Heude [18], Loreto Santa-Marina[28,36,37], Mariona Bustamante [28,32,33], Marie-France Hivert [2,4,38,41] & Jose Ramon Bilbao [1,39,41✉]

Higher maternal pre-pregnancy body mass index (ppBMI) is associated with increased neonatal morbidity, as well as with pregnancy complications and metabolic outcomes in offspring later in life. The placenta is a key organ in fetal development and has been proposed to act as a mediator between the mother and different health outcomes in children. The overall aim of the present work is to investigate the association of ppBMI with epigenome-wide placental DNA methylation (DNAm) in 10 studies from the PACE consortium, amounting to 2631 mother-child pairs. We identify 27 CpG sites at which we observe placental DNAm variations of up to 2.0% per 10 ppBMI-unit. The CpGs that are differentially methylated in placenta do not overlap with CpGs identified in previous studies in cord blood DNAm related to ppBMI. Many of the identified CpGs are located in open sea regions, are often close to obesity-related genes such as *GPX1* and *LGR4* and altogether, are enriched in cancer and oxidative stress pathways. Our findings suggest that placental DNAm could be one of the mechanisms by which maternal obesity is associated with metabolic health outcomes in newborns and children, although further studies will be needed in order to corroborate these findings.

A full list of author affiliations appears at the end of the paper.

Higher maternal pre-pregnancy body mass index (ppBMI) is associated with aberrant fetal growth[1], macrosomia and increased neonatal morbidity and mortality[2], and also with pregnancy complications such as pre-eclampsia, gestational diabetes, gestational hypertension, pre-term delivery and cesarean section[3]. It has been shown that maternal adipokine and insulin signaling in the placenta could contribute to regulate both the vascular development of this organ and the nutrient transport to the fetus, and therefore impact fetal development[3]. Additionally, it has also been observed that maternal ppBMI is associated with other offspring health outcomes in later life, including increased risk for obesity in children[4]. Observational studies have suggested links between maternal obesity and long-term risk of coronary heart disease, stroke, type 2 diabetes and asthma in offspring[5]. Very high maternal ppBMI has also been associated with poorer cognitive performance in children and greater risk of neurodevelopmental disorders[6], while there is also preliminary evidence in favor of potential implications in immune and infectious disease-related outcomes[3]. These associations could be mediated by epigenetic changes, including DNA methylation (DNAm), but the implication of an early epigenetic reprogramming *in utero* deserves further research[7].

A previous study carried out within the Pregnancy and Childhood Epigenetics (PACE) consortium[8] has shown that maternal ppBMI is widely associated with differences in cord blood DNAm in the newborn[9]. However, the authors observed that many of the significant epigenetic effects were modest (<0.2% methylation per BMI unit) and they did not detect enrichment for any particular biological pathway, leaving open questions regarding potential intra-uterine mechanisms that could be affecting the epigenetic profile of the newborn[9]. In this context, while the epigenetic alterations in cord and peripheral blood have been thoroughly investigated[9,10], the potential impact of maternal ppBMI in placental DNAm remains poorly explored. As far as we know, the most recent studies have performed methylation profiling with methylation arrays or reduced representation bisulfite sequencing in up to 300 term placentas of obese and non-obese mothers[11,12]. Although interesting, these studies have yielded a limited number of significant results, probably because of their relatively small sample size.

A recently published meta-analysis with 1700 placental samples by the PACE consortium[13] has discovered a placental DNAm signature of maternal smoking during pregnancy that is quite different from what has been observed in cord blood[14]. Differentially methylated CpGs related to smoking in pregnancy fall within active regions of the placental epigenome, and nearby genes are involved in the response to environmental stressors, regulation of inflammatory activity, and growth factor signaling. The placenta is a transient organ at the maternal-fetal interface, with endocrine and substrate-transport functions, that is sensitive to pregnancy environmental influences – exogenous or endogenous. Maternal pre-pregnancy obesity is often characterized by an adverse metabolic milieu that may alter placental function by increasing oxidative stress, vascular endothelium thickening, and inflammatory lesions in placental tissues[11]. Altogether, these facts encourage the investigation of the placenta as a putative mediator of maternal obesity and health outcomes in the offspring, specifically through the modification of the placental DNAm landscape.

In this context, the overall aim of the current analyses was to investigate the association of maternal ppBMI with epigenome-wide placental DNAm in 10 studies from the PACE consortium amounting to 2631 mother-child pairs. We also conducted functional enrichment analyses and comparison of our results with maternal ppBMI-associated cord blood DNAm alterations previously reported by PACE.

**Table 1 Effective sample size of the participating cohorts.**

| Cohort | Country | N | Array |
|--------|---------|-----|-------|
| AQUA | Australia | 95 | 450 K |
| EARLI | USA | 54 | 450 K |
| EDEN | France | 664 | 450 K |
| HBEC | USA | 186 | 450 K |
| Gen3G | Canada | 448 | EPIC |
| GENEIDA | Spain | 103 | 450 K |
| INMA | Spain | 168 | 450 K |
| ITU | Finland | 352 | EPIC |
| NHBCS | USA | 311 | 450 K |
| RICHS | USA | 250 | 450 K |
| TOTAL | | 2631 | |

## Results

**Study population.** Eleven North-American, Australian, and European studies ($N = 2631$) contributed to the epigenome-wide association study (EWAS) to determine the associations of maternal ppBMI with placental DNAm (Table 1), including Asking Questions about Alcohol in pregnancy (AQUA),[15] Early Autism Risk Longitudinal Investigation (EARLI)[16], Study on the pre- and early postnatal determinants of child health and development (EDEN)[17], Genetics of Glucose regulation in Gestation and Growth (Gen3G)[18], Genetics, Early Life Environmental Exposures and Infant Development in Andalusia (GENEIDA)[19], Harvard Epigenetic Birth cohort (HEBC)[20], Environment and Childhood Project (INMA)[21], The Intrauterine Sampling in Early Pregnancy Study (ITU)[22], Markers of Autism Risk in Babies-Learning Early Signs (MARBLES)[23], New Hampshire Birth Cohort Study (NHBCS)[24], and Rhode Island Child Health Study (RICHS)[25]. MARBLES was excluded at a later stage as the sample size was too small and results were inconsistent with the other cohorts (Supplementary Fig. 1).

Maternal ppBMI ($kg/m^2$) was generally self-reported. In those cases where ppBMI was not available, BMI in early pregnancy (1st trimester) was used. For simplicity, we will refer to both of them as maternal ppBMI. In all the analyses performed, we used ppBMI as a continuous variable. The cohort-specific average maternal ppBMI ranged from 22.9 in EDEN (France) to 27.6 in EARLI (USA) (standard deviation-SD = 1.60). In general, 548 (20.8%) and 369 (14.0%) mothers reported overweight (>25 BMI) and obesity (>30 BMI), respectively, while 115 (4.37%) appeared to be underweight (<18.5 BMI). Mean age of the mothers was 30.7 years (SD = 2.9). The distributions of other covariates by cohort are provided in Supplementary Data 1.

**Genome-wide DNAm meta-analyses.** Each cohort analyst conducted two different EWAS, modeling DNAm beta-values at a maximum number of 419,460 CpG sites in relation to maternal ppBMI using robust linear regressions, with and without adjustment for putative cellular components. Cell composition was estimated using the reference-free deconvolution algorithm RefFreeCellMix[26]. The CpGs included in the analyses were among those shared by the two most common Illumina Infinium Beadchip arrays, 450 K and EPIC, since both arrays were used to assess the DNAm levels of the samples, as shown in Table 1. All models were adjusted for maternal age, parity, maternal education and maternal smoking. Genomic inflation factors from the cohort-specific models (ranging from $\lambda = 0.692$ to 1.472) and a summary of the results can be seen in Supplementary Data 2. Finally, after quality control of the results, we conducted an inverse variance-weighted fixed-effects meta-analysis using the software GWAMA[27]. The inflation factors from the meta-analyses ($\lambda = 1.230$ and 1.220 for the cell type-adjusted and

-unadjusted models, respectively) (Fig. 1) revealed potential residual confounding and moderate inflation of test statistics.

After applying the Bonferroni correction for multiple-testing (meta-analysis nominal $p$ value <1.2e-07), we obtained 27 and 42 CpGs at which maternal ppBMI was significantly associated with placental DNAm in the models adjusted and unadjusted for cell type proportions, respectively. Full results for both models are provided in Supplementary Data 3 and 4, respectively. Higher maternal ppBMI was associated with lower placental DNAm in 24/27 differentially methylated CpGs identified in the cell type-adjusted model, while in the unadjusted model, 33/42 hits showed positive associations (higher maternal ppBMI associated with increased placental DNAm at the identified CpGs). However, beta-coefficients of CpGs that were differentially methylated in one model were positively correlated to the beta-coefficients of the same position in the other (Supplementary Fig. 2). Finally, the heterogeneity of associations across cohorts was lower for the model adjusted for cell type proportions compared to the unadjusted model (26/27 vs. 34/42 CpGs presented Cochran's Q-test $p$-values > 0.01) and thus, we continued with the results from the fully adjusted model for all downstream analyses.

Among the 27 differentially methylated CpGs identified in our cell type-adjusted EWAS (Table 2, Fig. 2A, B), a few individual CpGs are worthy of mention. The most notable association was observed at cg08219219, located in the eighth exon of EPHX3, with the lowest $p$-value in the meta-analysis (Bonferroni-corrected, meta-analysis $p$-value = 2.12e-05) and a beta-coefficient of −1.12e-03, meaning that a 10-unit difference in maternal ppBMI is associated with a 1.1% lower DNAm at this specific CpG site. This association was consistent across all cohorts (Cochran's Q-test $p$-value = 0.12) (Fig. 3a). The largest beta-coefficient was observed in cg14704941, in the first intron of CSRP3, with a positive beta-coefficient of 1.96e-03, corresponding to a 2% higher placental DNAm per 10-unit ppBMI (Bonferroni-corrected $p$-value = 2.24e-04 and Cochran's Q-test $p$-value = 0.09) (Fig. 3b). In turn, the largest negative beta-coefficient was found in cg04724807 (more than 57 Kb upstream of SYT16) with 1.8% lower DNAm per 10-unit ppBMI (Bonferroni-corrected $p$-value = 3.83e-04 and Cochran's Q-test $p$-value = 0.097) (Fig. 3c). The following CpGs reached the Bonferroni significance threshold and were not identified as highly heterogeneous across cohorts: cg00423969 and cg14163484, 1.5 kb upstream of the FER1L5 promoter, as well as cg26433445, cg15933729 and

cg08539067, close to CMIP, LGR4 and GPX1, respectively (Fig. 3d–h). The remaining Bonferroni-significant hits and their heterogeneity across cohorts are shown in Supplementary Fig. 3.

**Gene-set and regulatory enrichment analyses**. To gain insight into the biological processes that may be captured by placental DNAm associated with maternal ppBMI, we performed gene-set and regulatory enrichment analyses. To this end, first, we annotated CpGs to genes and regulatory elements as explained in the Material and Methods section. Then, we conducted gene-set enrichments for the 26 unique genes annotated to the 27 maternal ppBMI-sensitive CpGs with ConsensusPathDB[28] using KEGG, Reactome, Wikipathways and Biocarta reference databases. Two gene-set pathways were significantly enriched ($q$ value < 0.05), namely small cell lung cancer and oxidative stress-induced signaling pathway (Supplementary Data 5). This was also true when we reduced the background from default to only the genes that are represented in the Illumina 450 K array (~21,231). We also tested whether the genes annotated to maternal ppBMI-associated CpGs were enriched for regulatory regions of specific transcription factors (TFs). Most notably, our ppBMI-associated CpGs were enriched for genes regulated by ZNF217 (adjusted $p$ value = 0.02).

We then examined whether the 27 maternal ppBMI-associated CpGs were enriched for CpG island locations, placenta-specific imprinting regions or parent-of-origin-specific germline differentially methylated regions[29], regulatory features from the placenta-specific 15-chromatin state annotations from ROADMAP[30,31], or placenta-specific partially methylated domains[32] that contain placenta-specific repressed genes. We did not find any significant enrichment except for CpG island location and features: the maternal ppBMI-associated CpGs were depleted in CpG islands ($\chi^2 = -2.927$, $p$ value = 8.4e-04) and highly enriched in open sea regions ($\chi^2 = 2.742$, $p$ value = 1.3e-03) (Supplementary Fig. 4).

**Proximity to genetic variants relevant for birth outcomes**. We wanted to determine whether the maternal ppBMI-associated CpGs that we identified here were localized near genetic variants that have been associated with birth outcomes in previously published genome-wide association studies (GWAS). Thus, we investigated whether ppBMI-associated CpGs were within ± 0.5 Mb (1 Mb window) of single nucleotide polymorphisms (SNPs) that have been associated with birth weight (BW,

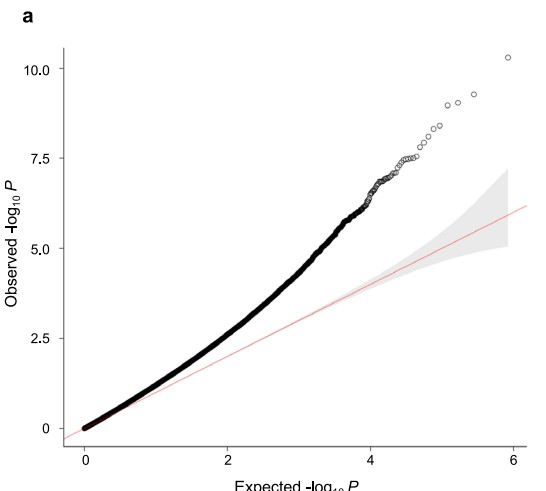
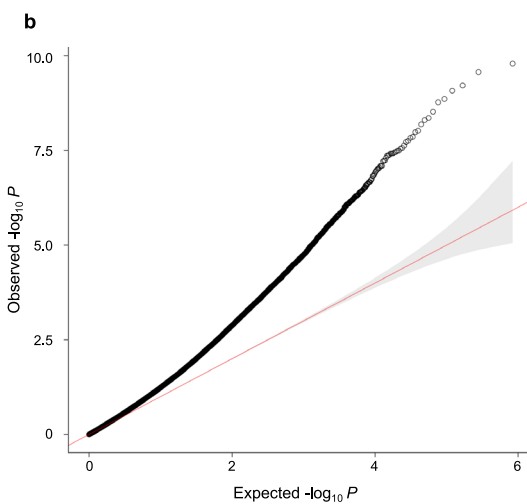

**Fig. 1 QQ-plots of the meta-analyses of the association between maternal ppBMI and placental DNAm.** Analyses were carried out **a** adjusting for putative cellular heterogeneity or **b** without adjusting for putative cellular heterogeneity ($N = 2631$ placental DNA samples). The red line represents the normal distribution and 95% confidence interval (gray shading).

**Table 2 Meta-analysis results from the model adjusted for maternal age, parity, maternal education, maternal smoking and putative cellular heterogeneity, showing the 27 Bonferroni significant CpGs associated with maternal ppBMI.**

| CpG | Results from the meta-analysis | | | | | Meta-analysis heterogeneity | | | | Location Annotations | | | | |
|---|---|---|---|---|---|---|---|---|---|---|---|---|---|---|
| | Beta | SE | 95% CI | P-value | Adj. P-value | Direction | HetISq | HetDf | HetPVal | Chr | Pos | Closest Gene | Distance to closest Gene | Relation to Island |
| cg00423969 | −0.00092 | 0.00015 | −0.0012;−0.0006 | 1.07E-09 | 4.49E-04 | −−−−−+−−−− | 0.474 | 10 | 0.047 | 2 | 97359879 | FER1L5 | 0 | Open Sea |
| cg00510149 | −0.00134 | 0.00025 | −0.0018;−0.0008 | 9.56E-08 | 4.01E-02 | −−+−−−−− | 0.531 | 10 | 0.023 | 11 | 10674966 | IRAG1 | 0 | Open Sea |
| cg02286857 | −0.00134 | 0.00025 | −0.0018;−0.0008 | 1.11E-07 | 4.66E-02 | −−−−−−−? | 0.602 | 9 | 0.010 | 2 | 47297177 | TTC7A | 0 | Open Sea |
| cg03603866 | 0.00129 | 0.00023 | 0.0008; 0.0017 | 1.16E-08 | 4.87E-03 | +−+−++?+ ++ | 0.167 | 9 | 0.294 | 13 | 107027625 | LINC00460 | 1284 | Open Sea |
| cg04724807 | −0.00179 | 0.00029 | −0.0024;−0.0012 | 9.12E-10 | 3.83E-04 | −+−−−−−− | 0.391 | 10 | 0.097 | 14 | 62396305 | SYT16 | 57496 | Open Sea |
| cg05590755 | −0.00080 | 0.00015 | −0.0011;−0.0005 | 1.19E-07 | 4.99E-02 | −+−−−−−− | 0.439 | 10 | 0.066 | 1 | 23855149 | E2F2 | 0 | North Shore |
| cg05965490 | −0.00132 | 0.00025 | −0.0018;−0.0008 | 1.16E-07 | 4.87E-02 | −−+−−−−− | 0.398 | 10 | 0.092 | 8 | 30264627 | RBPMS | 0 | Open Sea |
| cg08129759 | −0.00077 | 0.00014 | −0.0010;−0.0005 | 8.09E-08 | 3.39E-02 | −−−−−−−− | 0.303 | 10 | 0.166 | 1 | 202091944 | GPR37L1 | 83 | Open Sea |
| cg08219219 | −0.00112 | 0.00017 | −0.0014;−0.0008 | 5.05E-11 | 2.12E-05 | −−−−−−−− | 0.361 | 10 | 0.119 | 19 | 15337971 | EPHX3 | 0 | South Shelf |
| cg08539067 | −0.00049 | 0.00009 | −0.0007;−0.0003 | 2.84E-08 | 1.19E-02 | −−+−−−−+ | 0.491 | 10 | 0.039 | 3 | 49395985 | GPX1 | 193 | South Shore |
| cg09126859 | −0.00106 | 0.00019 | −0.0014;−0.0007 | 3.17E-08 | 1.33E-02 | −−+−−−−− | 0.415 | 10 | 0.081 | 12 | 52244063 | FIGNL2 | 18361 | South Shore |
| cg09167414 | −0.00152 | 0.00028 | −0.0021;−0.0010 | 3.31E-08 | 1.39E-02 | −−−−−−−− | 0.000 | 10 | 0.511 | 1 | 16076206 | TMEM82 | 1913 | South Shelf |
| cg12613632 | −0.00088 | 0.00016 | −0.0012;−0.0006 | 3.33E-08 | 1.40E-02 | −−−−−−+−− | 0.165 | 10 | 0.292 | 1 | 95385935 | CNN3 | 0 | Open Sea |
| cg14051770 | −0.00162 | 0.00029 | −0.0022;−0.0010 | 3.58E-08 | 1.50E-02 | −−−−−−−− | 0.000 | 10 | 0.833 | 7 | 76054572 | ZP3 | 0 | Open Sea |
| cg14143441 | −0.00174 | 0.00031 | −0.0023;−0.0011 | 1.55E-08 | 6.50E-03 | −−+−−−− | 0.513 | 10 | 0.030 | 8 | 134387493 | NDRG1 | 77945 | Open Sea |
| cg14163484 | −0.00085 | 0.00015 | −0.0012;−0.0005 | 4.13E-08 | 1.73E-02 | −−−?−−?−− | 0.224 | 8 | 0.251 | 2 | 97359926 | FER1L5 | 0 | Open Sea |
| cg14244402 | −0.00170 | 0.00032 | −0.0023;−0.0011 | 1.10E-07 | 4.61E-02 | −−−−−−−− | 0.122 | 10 | 0.330 | 9 | 130681102 | ST6GALNAC4 | 1796 | South Shore |
| cg14704941 | 0.00196 | 0.00032 | 0.0013; 0.0026 | 5.33E-10 | 2.24E-04 | ++++++++ +?.? | 0.432 | 8 | 0.090 | 11 | 19224659 | CSRP3 | 0 | Open Sea |
| cg15258080 | −0.00129 | 0.00022 | −0.0017;−0.0008 | 7.91E-09 | 3.32E-03 | −−−−−−−− | 0.419 | 10 | 0.078 | 10 | 71091204 | HK1 | 0 | Open Sea |
| cg15933729 | −0.00079 | 0.00014 | −0.0011;−0.0005 | 4.84E-09 | 2.03E-03 | −+−−−−− | 0.182 | 10 | 0.276 | 11 | 27504612 | LGR4 | 10277 | Open Sea |
| cg16310415 | 0.00172 | 0.00032 | 0.0011; 0.0023 | 4.99E-08 | 2.09E-02 | +++++−+ +−+? | 0.550 | 9 | 0.023 | 8 | 25898539 | PPP2R2A | 0 | North Shore |
| cg16724070 | −0.00163 | 0.00030 | −0.0022;−0.0010 | 8.22E-08 | 3.45E-02 | −+−−−−+−−? | 0.605 | 9 | 0.009 | 1 | 183313736 | LAMC1 | 19008 | Open Sea |

**Table 2 (continued)**

| CpG | Results from the meta-analysis | | | | | Meta-analysis heterogeneity | | | | Location Annotations | | | | |
|---|---|---|---|---|---|---|---|---|---|---|---|---|---|---|
| | Beta | SE | 95% CI | P-value | Adj. P-value | Direction | HetISq | HetDf | HetPVal | Chr | Pos | Closest Gene | Distance to closest Gene | Relation to Island |
| cg20042798 | −0.00052 | 0.00010 | −0.0007;−0.0003 | 7.95E-08 | 3.33E-02 | ------------ | 0.000 | 10 | 0.827 | 3 | 13036713 | IQSEC1 | 0 | South Shore |
| cg22673972 | −0.00085 | 0.00015 | −0.0011;−0.0005 | 3.12E-08 | 1.31E-02 | ------------ | 0.000 | 10 | 0.949 | 3 | 14415346 | SLC6A6 | 28728 | Open Sea |
| cg23696550 | −0.00131 | 0.00024 | −0.0018;−0.0008 | 5.80E-08 | 2.43E-02 | ------+------ | 0.136 | 10 | 0.318 | 14 | 24732386 | TGM1 | 0 | Open Sea |
| cg24893073 | −0.00061 | 0.00012 | −0.0008;−0.0004 | 1.04E-07 | 4.36E-02 | ------------ | 0.000 | 10 | 0.845 | 17 | 7742126 | KDM6B | 1107 | N_Shelf |
| cg26433445 | −0.00086 | 0.00015 | −0.0011;−0.0006 | 3.92E-09 | 1.64E-03 | ------------ | 0.571 | 10 | 0.013 | 16 | 81764289 | CMIP | 18921 | Open Sea |

Beta: effect size, SE: Standard error, 95%CI: 95% confidence interval, Adjusted P-value: Bonferroni corrected p-value, HetISq: Heterogeneity test statistic I², HetDf: Heterogeneity degrees of freedom, HetPVal: Heterogeneity test p-value. Direction: direction of the effect in each cohort, ordered in alphabetical order (+: positive; -: negative; and ?: CpG not present in the cohort).
CpGs that were not annotated with a gene name in the Illumina 450 K annotation file have been annotated with their closest gene.

$N = 310$), birth length ($N = 5$), head circumference ($N = 3$), gestational age (GA, $N = 6$) and BW + GA ($N = 6$)[33–38]. Of the total 330 birth outcome SNPs in autosomal chromosomes, 10 BW-associated SNPs were within 0.5 Mb of CpGs that were associated with maternal ppBMI. Therefore, more than a third of the 27 ppBMI-associated CpG sites were within 0.5 Mb of BW SNPs, including cg00423969 and cg14163484 (*FER1L5*), cg00510149 (*IRAG1*), cg02286857 (*TTC7A*), cg15258080 (*HK1*), cg22673972 (*SLC6A6*) and cg24893073 (*KDM6B*) (Supplementary Data 6).

**Comparison with maternal ppBMI-associated CpGs in cord blood DNAm.** We assessed whether the DNAm signatures of maternal ppBMI in the placenta were consistent with associations in cord blood previously reported by the PACE consortium[9]. We did not find any overlapping CpGs associated with maternal ppBMI between the two tissues. However, we reported three maternal ppBMI-associated CpGs in the placenta that were less than 0.5 Mb upstream or downstream from CpGs that had been associated with maternal ppBMI in cord blood: two out of the 3 *loci* identified showed consistent effect directions of the association with maternal ppBMI in both tissues (Supplementary Data 7).

**Discussion**
As far as we know, this is the largest EWAS meta-analysis conducted to date on placental DNAm. We have analyzed a total of 2631 mother-child pairs from 10 different PACE cohorts from Europe, America and Australia. We have identified 27 CpGs associated to maternal ppBMI, some of which showed up to 2% lower DNAm per 10-unit higher BMI. Although such a difference in BMI is unlikely in an individual woman in the context of pre-pregnancy interventions, we consider that it could represent the difference between women in the normal range of BMI and women with BMI in the obesity category.

The most significant association was observed for cg08219219, located in the eighth exon of *EPHX3*, for which a ppBMI difference of 10 units is associated with a 1.1% lower placental DNAm. It has been shown that soluble epoxide hydrolases such as *EPHX3* have higher activity in obese mice[39]. Additionally, it has been suggested that this family of hydrolases could act as therapeutic targets for metabolic and cardiovascular abnormalities related to obesity[40]. We highlighted two other significant hits showing the largest positive and negative beta-coefficients. cg14704941, in the first intron of *CSRP3*, presented with 2% higher placental DNAm per 10-unit ppBMI. *CRSP3* knockout mice develop dilated cardiomyopathy with hypertrophy and heart failure after birth[41]. The beta-coefficient of cg04724807, located upstream of *SYT16*, represented about 1.8% lower placental DNAm per 10-unit greater ppBMI. *SYT16* is over-expressed in pancreatic islet cells upon high glucose challenge and is thought to play a role in insulin secretion[42]. As previously stated, maternal obesity has been described to be associated with obesity, diabetes and cardiometabolic conditions in offspring later in life[4,5]. The fact that our EWAS identified CpGs near these metabolically relevant genes highlights the plausibility that they may play a role in the link between maternal obesity and future health outcomes in children.

Among our significant signals, we also found two CpG sites, cg00423969 and cg14163484, 1.5 kb upstream of the *FER1L5* promoter, presenting lower placental DNAm levels associated with higher maternal ppBMI. Remarkably, *FER1L5* encodes a dysferlin- and myoferlin-related protein, which has been predicted to have a role in vesicle trafficking and muscle membrane fusion events[43]. Both vesicle trafficking and membrane fusion are crucial events in placental development, since they allow the formation of the syncytiotrophoblast, an uninterrupted and

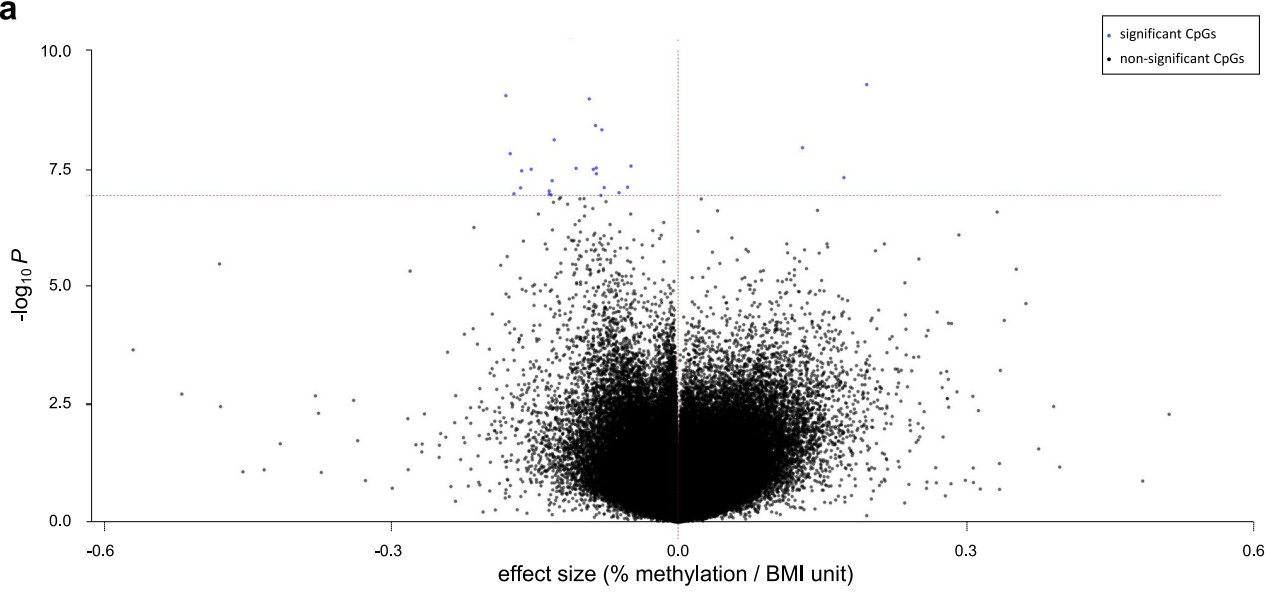

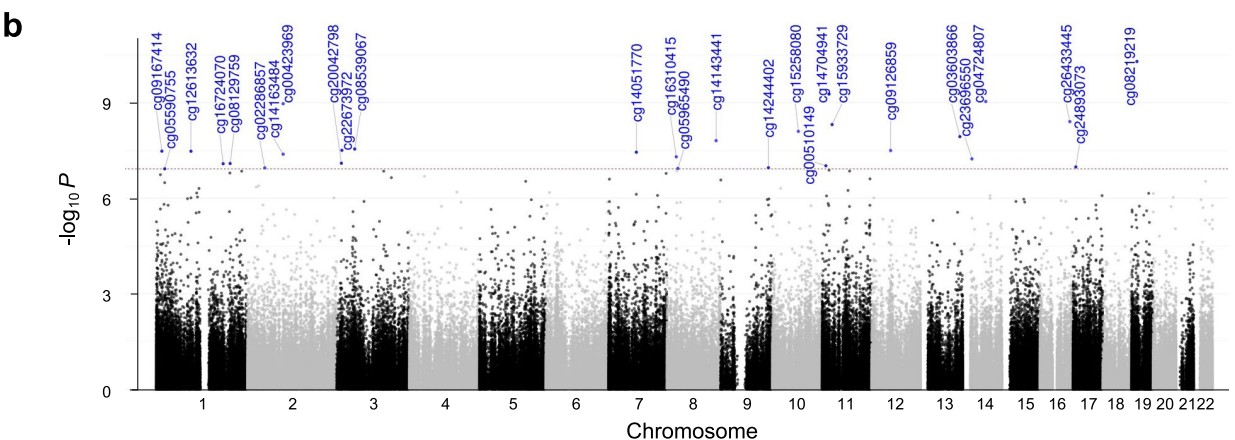

**Fig. 2 Association between maternal ppBMI and placental DNAm ($N = 2631$ placental DNA samples), after adjusting for maternal age, parity, maternal education, maternal smoking and putative cellular heterogeneity.** Association results are displayed as **a** volcano plot, where the X-axis shows the effect sizes (ranging between 0 and 1) in DNAm and **b** Manhattan plot, where the X-axis represents the genomic location of each CpG. In both panels blue dots indicate significantly associated CpGs (meta-analysis p-value < 1.2e-07).

multi-nucleated mass that covers the placental *villi* and enables the interplay with the mother[44]. In addition, *CMIP* and *GPX1*, two of the genes annotated to maternal ppBMI-associated CpGs, may present relevant biological roles in pregnancy. For example, different CpGs surrounding *CMIP* have been associated with preeclampsia in a placental DNAm study[45]. *GPX1* is an antioxidant gene and its mRNA levels are lower in the placenta of obese mothers compared to normal-weight mothers[46]. Finally, in the context of obesity, *LGR4*, another gene identified in the present study, bears an activating variant that contributes to abdominal visceral fat accumulation and therefore, to central obesity[47], suggesting that both genetic and epigenetic regulation at this locus may have a role in obesity-related phenotypes.

Regarding enrichment analyses, one of the most interesting findings is that several significant CpGs are located close to cancer-related genes. It has been recently described that the placenta is organized as a big mass of tumoral clones, with rapid cell divisions that enable selection for good cells that will eventually form the baby. Additionally, cancer and the syncytiotrophoblast of the placenta are both invasive tissues with many biological parallelisms[48,49]. Indeed, it is not surprising that factors that are relevant to the placenta, such as maternal obesity, could affect genes that are relevant to cancer. The other pathway that was enriched for altered genes is oxidative stress. It is well known that excessive fat mass accumulation is linked to oxidative stress. Moreover, peroxisomal fatty acid oxidation seems to be enhanced in the placenta of obese women, while mitochondrial activity is impaired, with a greater lipid storage and an altered transfer of lipids to the fetus[50]. Altogether, there is growing evidence suggesting that obesity-induced oxidative stress is a central factor involved in the risk for adverse outcomes in pregnancy[51,52].

Another interesting finding that deserves further investigation is the observation that differentially methylated CpGs are enriched for ZNF217 binding sites. This TF is epigenetically altered in placental cells under hypoxia[53], and it has been suggested that maternal obesity during pregnancy causes placental hypoxia[54]. However, whether this TF can drive the methylation machinery to selected regions of the genome and cause epigenetic changes

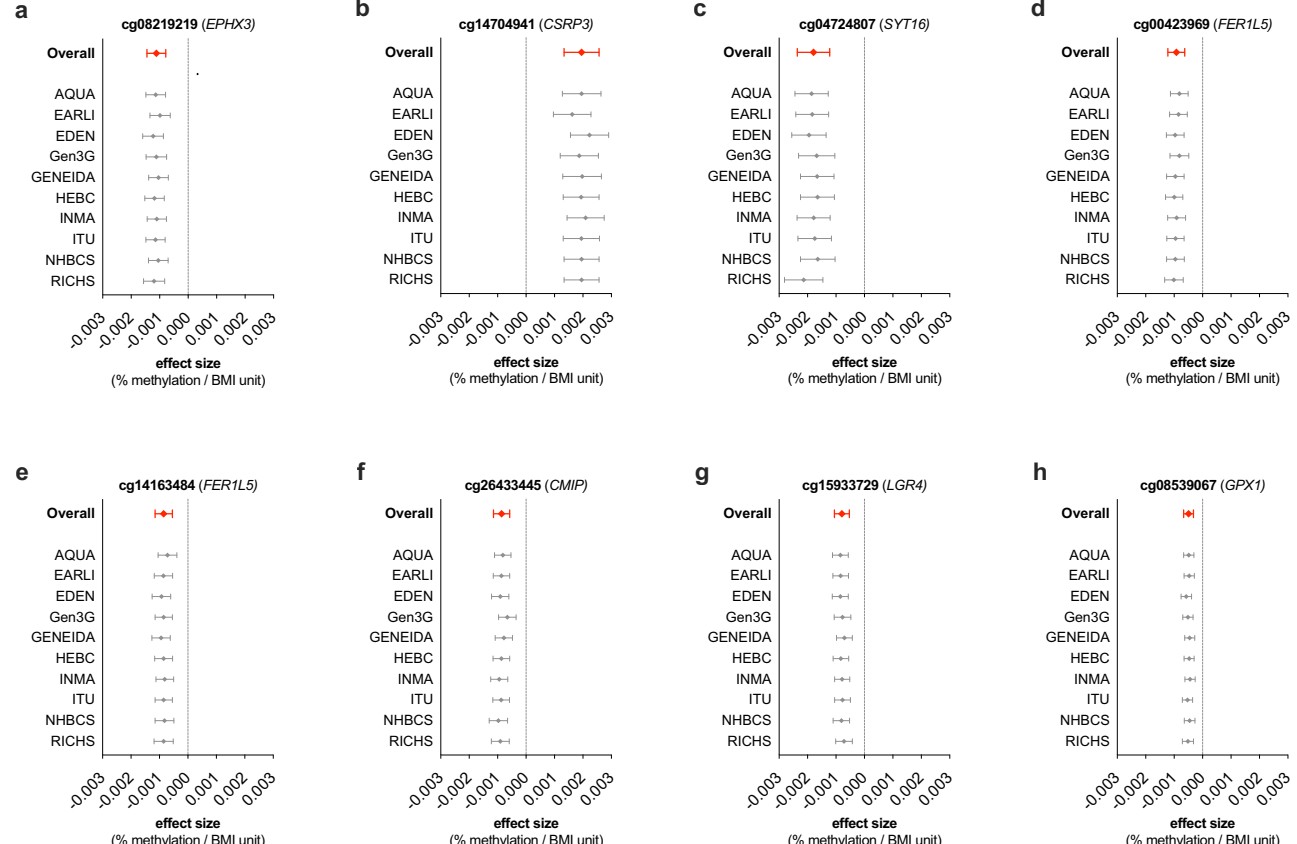

**Fig. 3 Forest plots of the leave-one-out analysis showing the fixed effects meta-analysis estimates of association between maternal ppBMI and placental DNAm.** Association of **a** cg08219219, **b** cg14704941, **c** cg04724807, **d** cg00423969, **e** cg14163484, **f** cg26433445, **g** cg15933729, **h** and cg08539067 with maternal ppBMI. In all panels, cohort names indicate the cohort excluded in each row, and error bars represent the 95% confidence interval of the effect size. Numerical source data for the figure are available in file Supplementary Data 9.

has not yet been explored. Similarly, the overlap between our CpGs and BW-associated regions suggests that both fetal genetic and placental epigenetic factors may contribute to the regulation of fetal growth, but this requires further research.

Our study has notable strengths but also several limitations. As previously mentioned, we have been able to coordinate a large number of cohorts and thus, to obtain an important sample size. Additionally, we have the experience of previous works, in which robust pipelines had already been implemented for EWAS, and we have run the quality control and meta-analysis in two independent institutions. Finally, none of the maternal ppBMI-associated CpGs from the current study are among the problematic probes with absolute methylation differences greater than 10% between Illumina 450 K and EPIC arrays that we identified in a previous study[55].

Regarding limitations, we did not have access to individual data addressing whether each ppBMI measurement was self-reported or taken at the end of the first trimester of pregnancy. Therefore, we cannot use this variable as a covariate nor compare between measurement types. We are very aware that self-reported ppBMI may not be the most accurate measurement for our variable of interest, as may also be the case of the measurement of BMI at the end of the first trimester. However, self-reporting of ppBMI has been shown to be reliable and highly correlated to measured BMI at 12 weeks of gestation ($r = 0.96$; $p$-value <0.0001)[56]. Second, the unavailability of genotype data in some of the participating cohorts did not allow to add genotype principal components to our models and there might be residual confounding by population structure that we did not account for. Third, most of the

cohorts were composed by a majority of individuals from European descent, which limits generalizability of our findings to other populations.

On the other hand, we are aware that RefFreeCellMix, the R package employed for adjustment of cell mixtures, is a principal component analysis-type correction method, and therefore presents the risk of over-correcting the results, especially in dense signal scenarios like the Illumina Beadchips, due to the capture of the signal by some of the top components of the estimation[57]. This, together with the fact that Bonferroni-correction is very strict and that our approach does not take into account the correlation between nearby CpGs, may have let some discoveries out of the focus. However, we have preferred to be strict and report only the most robust results.

In summary, here we present the largest EWAS of maternal ppBMI in association with placental DNAm performed to date. We identify 27 CpG sites at which we observe placental DNAm variations of 0.5–2.0% by 10-unit maternal ppBMI difference. Additionally, our DNAm findings seem to be placenta-specific, showing minimal overlap with a previous meta-analysis in cord-blood DNAm in relation to maternal ppBMI. The differentially methylated CpGs are mainly located in open sea regions, with a complete depletion from CpG islands, and enriched in cancer and oxidative stress- related pathways. These observations, together with the fact that maternal ppBMI is associated with placental DNAm at CpGs located close to obesity-related genes, leads us to hypothesize that placental DNAm could be one of the mechanisms by which maternal obesity is associated with aberrant fetal growth and maybe, other metabolic health outcomes in offspring

later in life. However, we cannot rule out that the changes observed could be markers of exposure to high ppBMI and therefore, our findings will need to be supplemented by functional studies or causal inference analyses to better understand if they truly have a role in pregnancy complications or long-term metabolic outcomes.

## Methods

**Participating cohorts**. Cohorts that are members of the PACE consortium, had existing DNAm data from placental tissue obtained with the Illumina 450 K or EPIC BeadChips, and had maternal BMI information prior to the beginning of pregnancy were invited to participate in the present study. The ten cohorts that contributed to the meta-analysis were AQUA, EARLI, EDEN, Gen3G, GENEIDA, HEBC, INMA, ITU, NHBCS and RICHS. All cohorts obtained ethics approval and informed consent from participants prior to data collection through their Institutional Ethics Boards. Exclusion criteria for this study were: non-singleton births, pre-eclampsia, and DNAm data not derived from the fetal facing side of the placenta. All participants included in this meta-analysis were of European ancestry. Detailed methods for each cohort are provided in Supplementary Note 1.

**DNAm data quality control and normalization**. All DNAm data processing and analyses were conducted in R 3.3.2[58], with the exception of the meta-analyses, which were performed with the GWAMA software (https://genomics.ut.ee/en/tools/gwama)[27]. DNAm from the fetal-facing side of the placenta was assessed with the Illumina 450 K or EPIC arrays. See Supplementary Note 1 for extra details on placenta collection, DNA extraction and DNAm acquisition in each cohort. In general, samples were randomized across the different arrays to avoid group differences derived from batch effects. Quality control of DNAm was standardized across all cohorts. Low-quality samples (showing a shifted beta-value distribution) were filtered out and probes with detection p-values >0.01 were excluded (for cohort-specific probe lists see Supplementary Data 8). DNAm beta-values were normalized with functional normalization[59] and beta-mixture quantile normalization (BMIQ)[60] was applied to correct for probe type bias. Cohorts examined their data for batch effects by depicting box-plots that divided the samples into different groups according to suspicious variables, and applied ComBat when applicable; all but one cohort (GENEIDA) identified batch effects and used ComBat to remove this source of variation. Probes that were exclusive for the EPIC array, hybridized to the X/Y chromosomes, cross-hybridizing probes and probes with SNPs at the CpG site, extension site, or within 10 bp of the extension site with an average minor allele frequency > 0.01 were filtered out[61]. Overall, 419,460 probes were available in the ten participating cohorts to assess placental DNAm. Methylation beta-values were modeled in robust linear regressions considering maternal ppBMI as a continuous variable. Finally, DNAm extreme outliers (<25th percentile - 3*IQR or >75th percentile + 3*IQR across all the samples) were trimmed.

**Estimates of putative cellular heterogeneity**. Putative cellular heterogeneity was estimated from DNAm data using a reference-free cell-mixture deconvolution method (RefFreeCellMix)[26]. The number of components varied between cohorts and ranged from 2 to 6, maybe because different sampling protocols result in differential heterogeneity across cohorts, or since the approach is data driven, those components could be capturing other major sources of variation in the array data, such as residual technical artifacts. Models for differential DNAm were corrected for the number of surrogate variables minus one to reduce multi-collinearity.

**Genome-wide differential DNAm analyses**. Within each cohort, robust linear regression from the MASS package[62] in R was run to account for potential heteroskedasticity and to test the associations between normalized placental DNAm beta-values at each CpG and maternal ppBMI. Models were adjusted for maternal age, parity, maternal education and maternal smoking during pregnancy. Cohorts ran models both with and without adjustment for RefFreeCellMix cell type proportions. Covariate data are described in more detail in Supplementary Note 1.

**Meta-analyses**. We performed inverse variance-weighted fixed effects meta-analyses using GWAMA[27]. The meta-analysis was performed independently by two groups to ensure consistent results and identical results were reproduced. We used the Bonferroni adjustment to correct for multiple testing. Secondary analyses were only performed on CpGs that passed the Bonferroni correction, particularly in the RefFreeCellMix-adjusted model. It is worth mentioning that both the meta-analysis and shadow meta-analysis teams performed a general quality control separately prior to the meta-analysis itself, showing consistent results. In summary, we checked that conflictive probes had been removed and drew the cohort-specific qq-plots, as well as the correlation between sample sizes and significant hits across cohorts. Finally, we checked the genomic inflation of the whole meta-analysis and plotted forest plots of the significant hits after leaving one cohort out at a time, to see whether any of the cohorts was guiding the associations. The MARBLES cohort

was excluded due to its small sample size and to the fact that forest plots showed inconsistencies compared to the rest of the cohorts (Supplementary Fig. 1).

**Functional and regulatory enrichment analyses**. We annotated CpGs to their closest genes and to CpG islands with annotations from the Illumina Human Methylation 450 K annotation file, and with several regulatory features using publicly available data: placental 15-chromatin states[30] released from the ROADMAP Epigenomics Mapping Consortium[31] (ChromHMM v1.10), placental germline differentially methylated regions[29] and placental partially methylated domains[32].

Over-representation analyses for gene-sets or pathways were performed at the gene level with ConsensusPathDB[28] using KEGG, Reactome, Wikipathways and Biocarta as reference databases. ConsensusPathDB performs a hypergeometric test, with a default background equal to the number of ConsensusPathDB entities that are annotated with an ID of the type the user has provided, and participate in at least one pathway. Finally, the program corrects multiple-testing with FDR. Enrichment for TFs was assessed at the gene level with EnrichR using ENCODE and ChEA consensus TFs from ChIP-X database. EnrichR results were ranked using the combined score (p-value computed using Fisher's exact test combined with the z-score of the deviation from the expected rank)[63].

**Overlap of ppBMI-sensitive CpG sites and birth outcome SNPs**. We assessed the genomic proximity between CpGs identified by our maternal ppBMI placental DNAm EWAS (Bonferroni significant in the cell-type adjusted model) and SNPs previously associated with BW, birth length, head circumference and GA[30–35]. Briefly, we verified the genomic proximity between SNPs from the largest GWAS performed to date on the above-mentioned birth outcomes and our identified CpGs by using the Genomic Ranges package[64] in R, within 1 Mb windows (±0.5 Mb) surrounding each of the 367 autosomal SNPs.

**Comparison of ppBMI-associated CpGs in placenta and in cord blood**. We examined whether maternal ppBMI-associated CpGs in placenta were the same as those previously reported in cord blood[9]. As no overlap was found between the hits that passed the Bonferroni correction in each study, we searched for CpGs from the cord blood study present 0.5 Mb upstream or downstream of each of the maternal ppBMI-associated CpGs in placenta (1 Mb windows), by using the GenomicRanges R package, with the aim of finding genomic regions where DNAm was related to ppBMI in the two different tissues.

**Statistics and reproducibility**. DNAm data from up to 419,460 CpG obtained with the Illumina 450 K or EPIC BeadChips in 2631 placental samples of the fetal side were normalized with functional normalization and BMIQ, and corrected for batch effects with ComBat if applicable. Afterwards, DNAm was correlated with maternal ppBMI in each of the participating cohorts using robust linear regressions with the MASS package in R 3.3.2. The code to perform the full analysis and the details are publicly available as described in the Code Availability section. The cohort-specific results were then meta-analyzed using the inverse variance-weighted fixed effects method in the GWAMA software, simultaneously in two independent laboratories. Results were fully consistent. Statistical significance was set at a meta-analysis nominal p-value = 1.2e-07, after Bonferroni correction for multiple-testing. Only CpGs below this threshold were taken into account in the downstream analyses. Generally, the statistical methods used in each of the downstream analyses were the ones suggested by the developers of each of the analytic tool implemented. More details in each of the analytical steps included in this study can be obtained in the specific section in Methods, Supplementary Note 1 and the code, as described in the Code Availability section.

**Reporting summary**. Further information on research design is available in the Nature Portfolio Reporting Summary linked to this article.

## Data availability

The full genome-wide results of the meta-analysis are presented in Supplementary Data 3 and 4 and have been uploaded to the EWAS Catalogue, available in Zenodo with identifier https://doi.org/10.5281/zenodo.7314534[65]. The individual level data used are not publicly available for several reasons. First, participants were not explicitly informed about this in the informed consent. Second, there are some studies that suggest that DNA methylation data has enough information to identify participants. Third, each PACE cohort follows different internal regulations in regards to public access of the data. Individual level data can still be shared with external researchers after signature of a data transfer agreement (DTA) with each of the participant cohorts, listed in Supplementary Note 1. More information is available in the PACE consortium website (https://www.niehs.nih.gov/research/atniehs/labs/epi/pi/genetics/pace/index.cfm). Source data underlying Fig. 3 is presented in Supplementary Data 9.

## Code availability

Scripts to reproduce the analysis have been deposited in a public GitHub repository and are available in Zenodo with identifier https://doi.org/10.5281/zenodo.7313966[66].

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

## Acknowledgements

We would like to thank the Pregnancy and Childhood Epigenetics (PACE) consortium, as well as all the families that participated in these studies for their generous contribution. This work was partially funded by GVSAN2018111086 from the Basque Department of Health and PI18/01142 from ISCIII - Spanish Ministry of Science and Innovation - cofounded by the ERDF "A way to make Europe" to JRB and LSM, respectively; and by the Joint Programming Initiative – A Healthy Diet for a Healthy Life (JPI HDHL) (NutriPROGRAM). ACP was supported by grant GVSAN2019111085 from the Basque Department of Health to NFJ. Detailed acknowledgements and funding for each participating cohort are described in Supplementary Note 1.

## Author contributions

N.F.-J., M.B., M.F.H. & J.R.B. conceived and designed the study. Study-specific analyses were completed by J. Loke. (AQUA), K.M.B. & J.F.D. (EARLI & MARBLES), A.F & B.H. (EDEN), R.F. (Gen3G), P.C.-S. & J.M.-M. (GENEIDA), A.M.B. (HEBC), N.F.-J. (INMA), T.K. (ITU), F.-Y.T. (NHBCS) and C.L. (RICHS). N.F.-J. & R.F. meta-analyzed the results. N.F-J. & A.C.-P. performed the follow-up analyses. N.F.- J., R.F., A.C.-P., M.F.H. & J.R.B. interpreted the results. N.F.-J., R.F., A.C.-P., M.F.H. & J.R.B. wrote the first draft of the manuscript. M.F.H. & J.R.B. contributed equally to guiding the direction of the project. All authors (N.F.-J., R.F., A.C.-P., J.L., P.P., T.K., F.-Y.T, C.L., A.M.B., M. Lozano., J-M.-M., Y.J.L., K.M.B., Y.Z., A.F., S.S., T.M.E., J.C., K.B.M., T.B., P.C.-S., J.H., M.D.F., J.M.L., J.T., D.C., M.F.F., A.G.-M., J.M.C., B.G.-A., R.J.S., J.F.D., E.M., M. Lacasaña., M.V., C.J.M., M.R.K., K.R., L.B., B.H., L.S.M., M.B., M.F.H. & J.R.B.) made substantial contributions to the acquisition, analysis, or interpretation of data, and read and critically revised the manuscript.

## Competing interests

The authors declare no competing interests.

## Additional information

[1]Department of Genetics, Physical Anthropology and Animal Physiology, University of the Basque Country (UPV/EHU) and Biocruces-Bizkaia Health Research Institute, Leioa, Basque Country, Spain. [2]Department of Population Medicine, Harvard Medical School and Harvard Pilgrim Health Care Institute, Boston, MA, USA. [3]University Grenoble Alpes, Inserm, CNRS, Team of Environmental Epidemiology Applied to Reproduction and Respiratory Health, IAB, Grenoble, France. [4]Centre de Recherche du Centre Hospitalier Universitaire de Sherbrooke (CHUS), Sherbrooke, QC, Canada. [5]Department of Psychology and Logopedics, University of Helsinki, Helsinki, Finland. [6]Gangarosa Department of Environmental Health, Rollins School of Public Health at Emory University, Atlanta, GA, USA. [7]Department of Environmental Medicine and Public Health, Icahn School of Medicine at Mount Sinai, New York, NY, USA. [8]Cancer Epidemiology Program, University of Hawaii Cancer Center, Honolulu, HI, USA. [9]Department of Epidemiology, Fielding School of Public Health, University of California, Los Angeles, CA, USA. [10]Epidemiology and Environmental Health Joint Research Unit, FISABIO-Universitat Jaume I-Universitat de València, Valencia, Spain. [11]Preventive Medicine and Public Health, Food Sciences, Toxicology and Forensic Medicine Department, Universitat de València, Valencia, Spain. [12]Department of Statistics and Operations Research, University of Granada, Granada, Spain. [13]Bioinformatics Unit. GENYO, Centre for Genomics and Oncological Research: Pfizer, University of Granada, Andalusian Regional Government, PTS Granada, Granada, Spain. [14]Murdoch Children's Research Institute, Parkville, VIC, Australia. [15]Department of Paediatrics, University of Melbourne, Parkville, VIC, Australia. [16]Department of Epidemiology, School of Public Health, University of Michigan, Ann Arbor, MI, USA. [17]Department of Medical Microbiology and Immunology, MIND Institute, Genome Center, University of California, Davis, CA, USA. [18]Université de Paris, Centre for Research in Epidemiology and Statistics (CRESS), INSERM, INRAE, Paris, France. [19]Department of Child and Adolescent Psychiatry and Psychology, Erasmus MC Rotterdam, The Netherlands. [20]Department of Epidemiology, Rollins School of Public health at Emory University, Atlanta, GA, USA. [21]Institute for Prevention and Cancer Epidemiology, Faculty of Medicine and Medical Center, University of Freiburg, Freiburg, Germany. [22]Health Research Institute of Asturias, ISPA and Biomedical Research and Innovation Institute of Cadiz (INiBICA), Research Unit, Puerta del Mar University Hospital, Cadiz, Spain. [23]Wendy Klag Center for Autism and Developmental Disabilities, Bloomberg School of Public Health, Johns Hopkins University, Baltimore, MD, USA. [24]Laboratory for Epigenetics & Environment, Centre National de Recherche en Génomique Humaine, CEA-Institut de Biologie François Jacob, Evry, France. [25]Max-Planck-Institute of Psychiatry, Department of Translational Research in Psychiatry, Munich, Germany. [26]University of Granada, Center for Biomedical Research (CIBM), Granada, Spain. [27]Instituto de Investigación Biosanitaria ibs.GRANADA, Granada, Spain. [28]CIBER of Epidemiology and Public Health (CIBERESP), Madrid, Spain. [29]Andalusian School of Public Health (EASP), Granada, Spain. [30]Deakin University, IMPACT – the Institute for Mental and Physical Health and Clinical Translation, School of Medicine, Geelong, Australia. [31]Department of Public Health Sciences and the MIND Institute, University of California Davis School of Medicine, Davis, CA, USA. [32]ISGlobal, Barcelona Institute for Global Health,

Barcelona, Spain. [33]Universitat Pompeu Fabra (UPF), Barcelona, Spain. [34]Department of Biochemistry and Functional Genomics, Universite de Sherbrooke, Sherbrooke, QC, Canada. [35]Department of Laboratory Medicine, CIUSSS du Saguenay–Lac-St-Jean – Hôpital Universitaire de Chicoutimi, Chicoutimi, QC, Canada. [36]Biodonostia, Epidemiology and Public Health Area, Environmental Epidemiology and Child Development Group, 20014 San Sebastian, Basque Country, Spain. [37]Health Department of Basque Government, Sub-directorate of Public Health of Gipuzkoa, San Sebastian, Basque Country, Spain. [38]Diabetes Unit, Massachusetts General Hospital, Boston, MA, USA. [39]CIBER of diabetes and associated metabolic disorders (CIBERDEM), Madrid, Spain. [40]These authors contributed equally: Nora Fernandez-Jimenez, Ruby Fore, Ariadna Cilleros-Portet. [41]These authors jointly supervised this work: Marie-France Hivert, Jose Ramon Bilbao. ✉email: joseramon.bilbao@ehu.eus

