## [Peer Review File · Communications Biology]

Reviewers' comments:

Reviewer #1 (Remarks to the Author):

Pre-pregnancy maternal body mass index and placental DNA methylation: a meta-analysis of 2,631 mother-child pairs

Fernandez-Jimenez et al. perform an epigenome wide association study to assess the relationship between DNA methylation and maternal pre-pregnancy BMI. While the experimental design and rationale are sound, I have some major concerns regarding the bioinformatic preprocessing and statistics. Additionally, with data and code not being available I cannot reproduce these findings myself. As such, I am unable to accurately review the results of this manuscript. Below are some of my concerns that need to be addressed prior to publication.

Major comments

1. Authors state that data were normalized with functional normalization and BMIQ. Were some samples normalized one way and others another? Were samples normalized with functional normalization and then BMIQ? What was the rationale for this. I think one normalization method should be used, otherwise authors may be introducing spurious batch effects based on normalization method.
2. Did authors use already preprocessed data from each cohort or did authors obtain raw data and preprocess themselves? I think the latter will ensure less batch effects in their own study and would yield more reproducible results.
3. Did the authors look at how samples were distributed on the array in each of the cohort. If groups are not distributed properly, using ComBat to correct for batch effects may actually introduce batch effects into the data. See Buhule et al. 2014. (<https://www.frontiersin.org/articles/10.3389/fgene.2014.00354/full>).
4. It would be helpful to have a table showing total probes to begin with and then how many probes were filtered out at each step.
5. Linear models should not be performed using beta values, and should be performed using M-values due to the heteroskedasticity in the data. See Du et al. 2010 (<https://link.springer.com/article/10.1186/1471-2105-11-587>) and Xie et al 2019 (<https://academic.oup.com/bioinformatics/article/35/7/1094/5089226?login=true>) for detailed explanation.
6. Trimming values outliers on the current dataset is biased. It takes out probes that may not be outliers in other datasets, limits the variance, and reduces the number of multiple test corrections needed, inflating the power of the analysis. If the authors want to remove outliers or probes that are likely not interesting, they should look for probes that are consistently outliers in multiple datasets that do not include their current data. I would suggest removing non variable probes. These can be found in Edgar et al. 2017 (<https://clinicaledgejournal.biomedcentral.com/articles/10.1186/s13148-017-0320-z>).
7. When performing enrichment analyses, what background did the authors use? The 450K is not representative of the entire genome, thus the background should be the genes/CpG sites present on the array.
8. Data and code is not publicly available. Code and data should be made available, otherwise it's impossible to reproduce the study. I understand there are ethical issues to providing data. If data cannot be made available there are a number of ways to provide population level statistics and simulated data along with code to allow reader to reproduce the analyses. Please see recent paper by Wilson et al 2021 (<https://febs.onlinelibrary.wiley.com/doi/full/10.1002/1873-3468.14067>) on how to properly share biological data and code.
 - a. Given that all these analyses are run in R, I am assuming there is no proprietary code, and this code is central to the conclusions of the paper and thus is required by Communications Bio to be made available.

Minor comments

1. Please cite R software and provide the version of R in which all analyses were performed.
2. How many placental samples were taken? Were placentas washed? Were membranes removed?
3. Authors state that quality control of DNA methylation was the same across all cohort. It would be helpful to describe briefly what quality control measures were taken in to account.
4. Figure 2a – X-axis is labeled as effect size but I cannot tell what the metric actually is.
5. Figure 2b – it would be helpful to have the data points above the FDR threshold labelled.
6. Text in all figure is small and difficult to read, please increase the text size.

Reviewer #2 (Remarks to the Author):

This is one of the largest studies looking at DNA methylation in the placenta and associations with maternal BMI. This type of analysis would not be possible without this approach, ie the combination of numerous cohorts. The authors identified methylation of several CpG sites being associated with maternal BMI. This is of great interest as changes in the placenta can impact fetal health in the short and long term. This is a great study which will be of great interest to those in the reproductive biology field as well as those interested in the developmental origins of disease. The methodological aspects of the paper will be of interest to all those in the epigenetics field. The comparison between placenta DNA methylation and cord blood methylation was also a good addition to the paper. The discussion of the limitations of the study was excellent. The methods were very robust. Future studies can now investigate whether the DNA methylation of the genes identified in this study are also altered at the transcriptional level in the placenta.

There are just some very minor things to be addressed. Please see below.

Methods

1. Spelling – “placental germline differentially methylated regions (gDRMs)²⁹” - should be gDMRs
2. Comparison of ppBMI-associated CpGs in placenta and in cord blood – was this a re-analysis? How was this performed?
3. Spelling error - “0.5 Mb upstream or downstream” should be downstream
4. Pre-eclampsia were excluded from this study but what about other pregnancy complications? These may have DNA methylation changes due to placenta pathology associated with pregnancy complications.

Results

1. Please check supplementary files as they are not labelled correctly (Supplementary Data 2 appears twice)

Pre-pregnancy maternal body mass index and placental DNA methylation: a meta-analysis of 2,631 mother-child pairs

RESPONSE TO REVIEWER #1:

Fernandez-Jimenez et al. perform an epigenome wide association study to assess the relationship between DNA methylation and maternal pre-pregnancy BMI. While the experimental design and rationale are sound, I have some major concerns regarding the bioinformatic preprocessing and statistics. Additionally, with data and code not being available I cannot reproduce these finding myself. As such, I am unable to accurately review the results of this manuscript. Below are some of my concerns that need to be addressed prior to publication.

We thank the reviewer for his/her suggestions and comments and apologize for not enabling the scripts from the very beginning. We have added them to the submission hoping that they will be of help for the reviewers. Additionally, they will be available upon request if the paper is finally accepted.

Major comments

1. Authors state that data were normalized with functional normalization and BMIQ. Were some sample normalized one way and others another? Were samples normalized with functional normalization and then BMIQ? What was the rationale for this. I think one normalization method should be used, otherwise authors may be introducing spurious batch effects based on normalization method.

Response: Both, functional normalization and BMIQ were used and this methodology was consistent across cohorts. We have now added the original R scripts used for the data processing to this submission (for the reviewers) and these scripts will be available upon request if the paper is finally accepted.

2. Did authors use already preprocessed data from each cohort or did authors obtain raw data and preprocess themselves? I think the latter will ensure less batch effects in their own study and would yield more reproducible results.

Response: Each cohort analyst preprocessed its cohort-specific data according to a common analytic plan provided by the main investigators leading this PACE consortium project. After running analyses according to the common analytic plan, cohort analysts provided the summary statistics to the meta-analysis analytic team. All the cohorts followed exactly the same scripts and instructions, and were guided across the analytical pipeline by the meta-analysis team. In conclusion, we do not expect major batch effects that would affect the meta-analyses. If there was any batch effects in one specific cohort, its influence would be minimal in the meta-analyses, and our meta-analytic QC steps and/or our sensitivity analyses would have revealed this cohort as an outlier (as in the case of MARBLES, that was excluded due to the fact that appeared to be an outlier in our leave-one-out analysis). In addition, the meta-analyses were run in parallel by another analytic team, reducing the risk of human errors at the meta-analyses' steps.

3. Did the authors look at how samples were distributed on the array in each of the cohort. Is groups are not distributed properly, using ComBat to correct for batch effects may actually introduce batch effects into the data. See Buhule et al. 2014. (<https://www.frontiersin.org/>).

*Response: Thank you for raising this very interesting point. In each cohort, samples were randomized across the different arrays to limit batch effect. Please see **Supplementary Methods** for details on how cohorts managed samples and potential batch effects at the analytic level (individually for each cohort). However, it would be very unlikely that samples organization on the arrays would bias our results, given that we used maternal BMI as a continuous variable (not as 'group' to compare high and low BMI), and that no cohorts selected samples based on maternal BMI. To be fully transparent, we added precisions regarding this point in the main text (see Methods section).*

4. It would be helpful to have a table showing total probes to begin with and then how many probes were filtered out at each step.

Response: We have added a datasheet (Supplementary_file_8.xlsx) with the probes removed at each step within each cohort.

5. Linear models should not be performed using beta values, and should be performed using M-values due to the heteroskedasticity in the data. See Du et al. 2010 (<https://link.springer.com/>) and Xie et al 2019 (<https://academic.oup.com/>) for detailed explanation.

Response: It is true that linear regression should be applied in normal data as M-values. We are also conscious that using linear regression in betas may decrease the statistical power in extreme methylation values (as pointed out in the references you provided). However, it is widely accepted to use this kind of methods in beta-values given that using betas yields much more biologically interpretable results. In this sense, we have chosen to reinforce the interpretability of the results although we are aware that this will probably cause that we miss some real associations. Additionally, in this particular study, we use the robust linear regression method of the MASS package, which has been shown to mitigate the effect of the heteroskedasticity of the data. This has been previously implemented in other papers of the PACE consortium such as a quite recent work published in Nat Commun (Everson et al, 2021; <https://www.ncbi.nlm.nih.gov/pmc/articles/PMC8384884/>). In conclusion, we understand the criticism by the reviewer but by using Robust Linear Regression we limit the issues related to the distribution of the methylation in beta-values, and avoid the interpretation problems that M-values would cause. This is clarified in the text.

Methods; Genome-wide differential DNAm analyses (first line)

“Within each cohort, robust linear regression from the MASS package⁶¹ in R was run to account for potential heteroskedasticity to test the associations between normalized placental DNAm beta-values at each CpG and maternal ppBMI.”

6. Trimming values outliers on the current dataset is biased. It takes out probes that may not be outliers in other datasets, limits the variance, and reduces the number of multiple test corrections needed, inflating the power of the analysis. If the authors want to remove outliers or probes that are likely not interesting, they should look for probes that are consistently outliers in multiple datasets that do not include their current data. I would suggest removing non variable probes. These can be found in Edgar et al. 2017 (https://).

Response: Each cohort trimmed the outliers at the individual level, so those probes are included in the meta-analysis unless they were outliers in all the independent cohorts. In this sense, we believe not to have altered the variance significantly. Regarding the non-variable probes, we have compared the less variable probes of some of our cohorts to those provided in Edgar’s list and they are not entirely coincident. We think that more efforts will be needed in order to find a consistent invariable probe list in placenta and other tissues.

7. When performing enrichment analyses, what background did the authors use? The 450K is not representative of the entire genome, thus the background should be the genes/CpG sites present on the array.

Response: In the over-representation analysis we performed, the user can provide a list of identifiers of interesting genes or proteins, e.g. of genes showing CpGs associated to a certain phenotype. The gene identifiers are mapped to physical entities in ConsensusPathDB. Over-represented sets are searched among three categories of predefined gene sets: network neighborhood-based sets, pathway-based sets and Gene Ontology-based sets. For each of the predefined sets, a p-value is calculated according to the

hypergeometric test based on the number of physical entities present in both the predefined set and user-specified list of physical entities. If no background is uploaded by the user (corresponding to the list of IDs of all measured entities in the experiment), the background parameter value for the hypergeometric test will depend on the type of the accession numbers used for the input list: more precisely, the background size is the number of ConsensusPathDB entities that are annotated with an ID of the type the user has provided, and participate in at least one pathway / GO category / neighborhood-based entity set (depending on which of these predefined classes are considered by the user). This will probably result in an even more stringent analysis compared to that in which the background is manually uploaded by the user. For more information see: <http://cpdb.molgen.mpg.de/CPDB/tutorial#pathwaya.wilcoxon>

However, reducing the universe list to the genes that are analyzed in the 450K array (~21,231 genes) and conserving the same parameters (q -value < 0.05, databases: KEGG, Reactome, Wikipathways, and Biocarta), we got the enrichment of the same pathways (oxidative stress and small cell lung cancer), plus cellular senescence pathway and responses to external stimuli:

select all none	pathway name	set size	candidates contained	p-value	q-value	pathway source
[ ]	Oxidative Stress Induced Senescence	125(57)	2 (3.5%)	0.00229	0.026	Reactome
[ ]	Small cell lung cancer - Homo sapiens (human)	92(92)	2 (2.2%)	0.00586	0.026	KEGG
[ ]	Small cell lung cancer	96(96)	2 (2.1%)	0.00636	0.026	Wikipathways
[ ]	Cellular Senescence	187(108)	2 (1.9%)	0.00799	0.026	Reactome
[ ]	Cellular responses to stress	554(447)	3 (0.7%)	0.0173	0.0407	Reactome
[ ]	Cellular responses to external stimuli	569(461)	3 (0.7%)	0.0188	0.0407	Reactome

In conclusion, we have added in the main text more details about our selected method and explained that results were consistent when the background was manually uploaded.

8. Data and code is not publicly available. Code and data should be made available, otherwise it's impossible to reproduce the study. I understand there are ethical issues to providing data. If data cannot be made available there are a number of ways to provide population levels statistics and simulated data along with code to allow reader to reproduce the analyses. Please see recent paper by Wilson et al 2021 (<https://febs.onlinelibrary>.) on how to properly share biological data and code.
a. Given that all these analyses are run in R, I am assuming there is no proprietary code, and this code is central to the conclusions of the paper and thus is required by Communications Bio to be made available.
Response: Once the paper is accepted, we will make the summary stats available and code will be available upon request. We can consider any other options upon editorial request.

Minor comments

1. Please cite R software and provide the version of R in which all analyses were performed.

Response: We have provided the details in the text.

2. How many placental samples were taken? Were placentas washed? Were membranes removed?

Response: Each cohort had their own respective protocol and provided descriptions of their respective process in Supplementary Methods. We are aware the heterogeneity of collection protocols across cohorts is a limitation of meta-analyses like ours but in general, placenta biopsies were collected at the time of delivery, washed with PBS and immediately stored in a way not expected to harm the DNA methylation patterns.

3. Authors state that quality control of DNA methylation was the same across all cohort. It would be helpful to describe briefly what quality control measures were taken in to account.

Response: We have tried to be more clear about the quality control steps in the revised version.

Methods; DNAm data quality control and normalization (line 5)

“In general, samples were randomized across the different arrays to avoid group differences derived from batch effects. Quality control of DNAm was standardized across all cohorts. Low quality samples (showing a shifted beta-value distribution) were filtered out and probes with detection p-values > 0.01 were excluded (for cohort-specific probe lists see Supplementary File 8). DNAm beta-values were normalized with functional normalization⁵⁸ and beta-mixture quantile normalization (BMIQ)⁵⁹ was applied to correct for probe type bias. Cohorts examined their data for batch effects by depicting boxplots that divided the samples into different groups according to suspicious variables, and applied ComBat when applicable; all but one cohort (GENEIDA) identified batch effects and used ComBat to remove this source of variation. Probes that were exclusive for the EPIC array, hybridized to the X/Y chromosomes, cross-hybridizing probes and probes with SNPs at the CpG site, extension site, or within 10 bp of the extension site with an average minor allele frequency > 0.01 were filtered out⁶⁰.”

On the other hand, we have added a paragraph to inform about other types of quality controls, performed by the meta-analyses teams prior to the meta-analysis.

Methods; Meta-analyses (line 5)

“It is worth of mention that both the meta-analysis and shadow meta-analysis teams performed a general quality control separately prior to the meta-analysis itself, showing consistent results. In summary, we checked that conflictive probes had been removed and drew the cohort-specific qq-plots, as well as the correlation between sample sizes and significant hits across cohorts. Finally, we checked the genomic inflation of the whole meta-analysis and plotted forest plots of the significant hits to see whether any of the cohorts was guiding the associations. The MARBLES cohort was excluded due to its small sample size and to the fact that forest plots showed inconsistencies compared to the rest of the cohorts.”

4. Figure 2a – X-axis is labeled as effect size but I cannot tell what the metric actually is.

Response: Thank you for pointing this out, we are sorry we had omitted this important information. The metric is % methylation/BMI unit and it has been added to the X-axis label.

5. Figure 2b – it would be helpful to have the data points above the FDR threshold labelled.

Response: We have modified the figure according to reviewer’s suggestion

6. Text in all figure is small and difficult to read, please increase the text size.

Response: We thank the reviewer for the suggestions regarding the figures. We believe that they look much better now. Figures have been redone following the reviewer’s suggestion.

RESPONSE TO REVIEWER #2:

This is one of the largest studies looking at DNA methylation in the placenta and associations with maternal BMI. This type of analysis would not be possible without this approach, ie the combination of numerous cohorts. The authors identified methylation of several CpG sites being associated with maternal BMI. This is of great interest as changes in the placenta can impact fetal health in the short and long term. This is a great study which will be of great interest to those in the reproductive biology field as well as those interested in the developmental origins of disease. The methodological aspects of the paper will be of interest to all those in the epigenetics field. The comparison between placenta DNA methylation and cord blood methylation was also a good addition to the paper. The discussion of the limitations of the study was excellent. The methods were very robust. Future studies can now investigate whether the DNA methylation of the genes identified in this study are also altered at the transcriptional level in the placenta.

We thank the reviewer for his/her very kind words and will take the comments into very serious consideration with the aim of improving the manuscript.

There are just some very minor things to be addressed. Please see below.

Methods

1. Spelling – “placental germline differentially methylated regions (gDRMs)29” - should be gDMRs

Response: done.

2. Comparison of ppBMI-associated CpGs in placenta and in cord blood – was this a re-analysis? How was this performed?

Response: Thank you for allowing us to clarify our approach. Our goal was to compare the ‘top’ hits obtained in the different studies. This is explained in the text (with several edits compared to the previous version that we believe will help clarify the procedure).

Methods; Comparison of ppBMI-associated CpGs in placenta and in cord blood (line 1)

“We examined whether maternal ppBMI-associated CpGs in placenta were the same as those previously reported in cord blood⁹. As no overlap was found between the hits that passed the Bonferroni correction in each respective study, we searched for CpGs from the cord blood study present 0.5 Mb upstream or downstream of each of the maternal ppBMI-associated CpGs in placenta (1 Mb windows), by using the GenomicRanges R package, with the aim of finding genomic regions where DNAm was related to ppBMI in the two different tissues.”

3. Spelling error - "0.5 Mb upstream or dowsstream" should be downstream - *Response: done.*

4. Pre-eclampsia were excluded from this study but what about other pregnancy complications? These may have DNA methylation changes due to placenta pathology associated with pregnancy complications.

Response: We appreciate this point, and we agree with the reviewer that many different pregnancy complications may influence the placenta DNAm pattern. However, not all cohorts had details of all possible complications in late pregnancy or at delivery, so we did not require studies to exclude samples based on other complications, given that it could lead to selection bias (excluding samples in some cohorts but still likely included in others if they don't have the information). Given that these samples likely represent a small number of the overall sample size, we feel it's unlikely that it would highly influence our results.

Results

1. Please check supplementary files as they are not labelled correctly (Supplementary Data 2 appears twice)

Response: We thank the reviewer for this comment. This mistake has been corrected.

Reviewers' comments:

Reviewer #1 (Remarks to the Author):

The authors made a few changes to the manuscript which I appreciate. However, the majority of my statistical concerns still hold. While it is common to see others perform linear regression on beta values, that does not mean it is statistically correct.

I am also concerned with performing BMIQ and functional normalization and what that could do to the data. Only one normalization is needed.

Additionally, having "data and code available upon request" is against the journal's policy. While I appreciate that there is likely privacy issues, raw files or private health information needed to reproduce the work can be put in access controllable repositories, such as EGA. Given that this is array data and will not contain individual level sequence data, IDATS can be put on GEO.

Reviewer #2 (Remarks to the Author):

All comments have been adequately addressed.

Pre-pregnancy maternal body mass index and placental DNA methylation: a meta-analysis of 2,631 mother-child pairs

Response to reviewers

Reviewer #1 (Remarks to the Author):

The authors made a few changes to the manuscript which I appreciate. However, the majority of my statistical concerns still hold. While it is common to see others perform linear regression on beta values, that does not mean it is statistically correct.

Response: The reviewer insists on the idea that we are using a linear regression but we are using a robust linear regression instead, which has been conceived precisely to avoid the problems that heteroskedasticity would cause. Additionally, one of the main papers mentioned across the literature to reinforce the idea of the use of M values over beta values (namely Du et al., BMC Bioinformatics 2010) is based on a methylation titration experiment in which the performance of beta and M values is compared by using linear regressions and Pearson coefficients. The authors state as the main conclusion that “the Beta-value has a more intuitive biological interpretation, but the M-value is more statistically valid for the differential analysis of methylation levels”. While this is true, we want to highlight that they do not use a robust linear regression as we do and therefore, we would like to make it clear that their conclusions cannot be extrapolated to our work. In fact, there are at least five general strategies for dealing with heteroscedasticity: the main two would be either to transform the data (as in the case of the M values) or to replace the least squares estimator with some type of robust estimator. (as we propose). In their article published in 2010 in Understanding Statistics from Taylor & Francis “Robust Regression Methods: Achieving Small Standard Errors When There Is Heteroscedasticity”, Rand R. Wilcox & H. J. Keselman stated that the latter appears to be the best current approach. This would be a better option than to transform the data because “in some instances transforming the data improves matters, but this is not always the case. A particular concern here is that generally these transformations do not deal effectively with outliers. More precisely, even after data have been transformed, outliers can remain.”

I am also concerned with performing BMIQ and functional normalization and what that could do to the data. Only one normalization is needed.

Response: Beta-mixture quantile (BMIQ) normalization corrects the bias associated with type 2 probe values into a distribution that is comparable with that of type 1 probes in 450K methylation microarrays. While BMIQ normalize the data within each sample, functional normalization is a between-sample normalization method. Thus, we consider the two type of normalization methods to be complementary. This kind of method combinations have been described previously, showing high sensitivity and consistency when using either Noob+BMIQ, Noob+SWAN or FunNorm+BMIQ (like in our work). For more details please see:

<https://bmcbioinformatics.biomedcentral.com/articles/10.1186/s12864-016-2819-7#Sec2>

Additionally, having "data and code available upon request" is against the journal's policy. While I appreciate that there is likely privacy issues, raw files or private health information needed to reproduce the work can be put in access controllable repositories, such as EGA. Given that this is array data and will not contain individual level sequence data, IDATS can be put on GEO.

Response: The full genome-wide results of the meta-analysis are publicly available as Supplementary Data 3 and 4. Additionally, they will be uploaded to the EWAS Catalogue once the article is accepted.

Scripts to reproduce the analysis can be found in a public github repository (<https://github.com/ariadnacilleros/EWAS-maternal-BMI-vs-placental-methylation>). The individual level data used are not publicly available for several reasons. First, participants were not explicitly informed about this in the informed consent. Second, there are some studies that suggest that DNA methylation data has enough information to identify participants. Third, each PACE cohort follows different internal regulations in regards to public access of the data. Nonetheless, individual level data can still be shared with external researchers after signature of a data transfer agreement (DTA). We have included these considerations in the manuscript.

Reviewers' comments:

Reviewer #3 (Remarks to the Author):

In this study, the authors investigate if associations between pre-pregnancy BMI (ppBMI) and various birth defects could be mediated by DNA methylation. Towards this, the authors make use of placenta methylation data from 2600 mother-child pairs from the PACE consortium.

While I cannot comment on the statistical analysis of this study, I find the presented results and conclusions are very limited. The authors find 27 CpGs to be significantly associated with ppBMI. The reported differences in methylation 0.5-2% per 10 BMI units and I wonder how much such minimal changes in methylation can have any influence on downstream processes.

The follow-up analysis to interpret associations of these CpGs with nearby genes and regulatory elements and the drawn conclusions from this data are not very convincing, given the small amount of CpGs used.

Furthermore, the authors state in the abstract: "Our findings suggest that placental DNAm could be one of the mechanisms by which maternal obesity is associated with metabolic health outcomes in newborns and children". I suggest to rephrase to "we find correlations between placental DNAm, maternal obesity is associated with metabolic health outcomes in newborns and children " since the original sentence is misleading and no experiments are provided to validate this claim.

Reviewer #4 (Remarks to the Author):

The authors present a beautifully written manuscript and response to the previous round of review. As the research question was on maternal pre-pregnancy body mass index, I was very heartened that the authors performed methylation analysis in placental tissue, rather than blood. The authors further compared results with prior findings obtained from cord blood. In summary, the authors identified 27 CpGs that were associated with maternal ppBMI at experiment-wide significance. The gene and functional annotations provide helpful biological insights into the determinants of maternal ppBMI.

With regards to the statistical treatment of the data, I have to humbly agree with the points brought up by Referee #1. I particularly appreciate this reference that was raised by the authors in their response: "Robust Regression Methods: Achieving Small Standard Errors When There Is Heteroscedasticity", Rand R. Wilcox & H. J. Keselman". However, in a genome-wide search, certain additional safeguards should also be considered in the light of the very high genomic inflation (of 1.23 and 1.22) in the meta-analysis. Such a high residual inflation is normally unacceptably high in genome-wide studies (see a particularly vigorous discussion here: Sawcer S et al., Genetic risk and a primary role for cell-mediated immune mechanisms in multiple sclerosis. Nature. 2011; 476(7359):214-9). In such instances within the context of genome-wide genetic screens, where a very large number of markers are tested, other alternate methods of analysis could be considered and explored. One of these methods is the linear mixed model (described in detail in Sawcer et al., 2011), and I pray the authors could consider it.

Archived: 24 June 2022 11:03:16

From: Jose Ramon Bilbao

Mail received time: Thu, 23 Jun 2022 09:24:40

Sent: Thu, 23 Jun 2022 09:24:30

To: Communications Biology Christina Rosenthal

Cc: Hivert, Marie-France, M.D. Mariona Bustamante ariadna cilleros portet Nora Fernandez-Jimenez

Subject: Re: Decision on manuscript COMMSBIO-21-2748B

Importance: Normal

Sensitivity: None

[External - Use Caution]

Dear Editor,

We are writing to you as the main authors of article COMMSBIO-21-2748B, entitled, "*Pre-pregnancy maternal body mass index and placental DNA methylation: a meta-analysis of 2,631 mother-child pairs*", to manifest our profound disappointment and our complete disagreement with the rejection. We would like to explain in detail the reasons why we feel that this decision is unfair and totally unsupported by facts and/or scientific arguments.

In the first place, we think that reviewer 3 and 4 have not provided any evidence, reference or argument to support the rejection of our paper.

Regarding the comments by reviewer 3, he/she considers that our findings are insignificant in terms of biological consequences. While the effect sizes reported are certainly modest, there are multiple examples of published EWAS that have identified very "*small*" DNAm differences in terms of effect size, but our investigations are actually carried out under the premise that these *small* changes are showing effects that might have long-term consequences in biological programming. Additionally, small changes can reflect larger effects in one or a few cell types, which are diluted in the bulk data. In fact, we wonder whether reviewer 3 has ever found any EWAS meta-analysis paper in which numerous big changes have been reported. This is highly unlikely and would definitely be suspicious of spurious association. Moreover, reviewer 3 claims not to have any comments about the statistics (in view of the previous review rounds, we can guess that that was precisely the purpose for which he/she was asked to review our paper in the first place). However, he/she does not specify whether this lack of comments is due to the fact that he/she is not an expert in the methodology, or because he/she agrees with the analysis pipeline we have used. In any case, the Editors knew about the impact of our findings (that we do not overclaim) from the beginning, when we were offered a transfer from Nat Commun. In conclusion, we think that the revision by reviewer 3 cannot be taken into consideration because: i) he/she just gives an opinion on the biological meaningfulness of our findings, ignoring the general nature of EWAS results; ii) this was already known by the Editors from the very beginning of the revision process and has previously been assessed by reviewers 1 and 2 and, iii) this opinion is completely out of focus, since we assume he/she was asked to review the statistics of the paper.

On the other hand, regarding the comments by reviewer 4, we do not understand how someone can advise to reject a paper without discussing any of the points previously raised (that again, we guess were the reason why this reviewer was contacted). He/she says that he/she "*humbly agrees*" with reviewer 1 but does not explain what exactly he/she agrees with. Reviewer 4 states, for example, that he/she appreciates one of the references that we propose to support the use of robust statistics instead of M values. We wonder how one can *appreciate* reference that refutes the opinion with which he/she says to "*humbly agree*". For the rest of the points raised by reviewer 1, he/she does not even write a word. We again wonder why using both BMIQ and functional normalization is a bad idea when most of the literature supports this approach and moreover, any work that does not perform both intra- and inter-sample normalization would probably be immediately rejected for that particular reason. Regarding

the publication of raw epigenomic datasets, we would also like to know which would be the ideal solution for reviewer 4 (because we guess that publishing sensitive epigenomic data without the consent of the participant mothers and children has not been suggested).

In general, about the statement by reviewer 4 "*With regards to the statistical treatment of the data, I have to humbly agree with the points brought up by Referee #1. I particularly appreciate this reference that was raised by the authors in their response*", we consider it is not a scientifically solid, reasoned statement. We want to highlight that this is the only phrase that refers to the points raised by reviewer 1, that we guess were the reason to contact reviewers 3 and 4, in the whole third round of revisions. The fact that we are rejected without any solid arguments against our rationale is even more painful than rejection itself. Accepting this statement of reviewer 4 as the main reason to reject our paper is against fair play and constructive scientific discussion.

Last but not least, the rest of the criticism by reviewer 4 reveals his/her own expertise on GWAS but it is, again and from our point of view, out of the focus for two reasons: i) the genomic inflation of our work has not been mentioned previously and should not be the main topic in the third round of revisions; however, ii) the statements by the reviewer could be interesting if our work was a GWAS, but it is an EWAS. In fact, while a genomic inflation of 1.2 is generally unacceptably high for GWAS (because the association with the trait is calculated with genotypes and thus, discrete values), it is considered to be quite moderate for EWAS, where the trait of study associates with methylation as a continuous variable. However, there are softwares to reduce the genomic inflation and we could easily run one of them and report that in the article, if this was a problem towards the acceptance of our work. If we have not done it before it is because we did not want to overcorrect the data (and were confident that our moderate genomic inflation was going to be considered acceptable by the majority of EWAS experts), and because we have not been asked to do so in the first two rounds of revisions.

Finally, we feel confused by the recommendation of the reviewer when he/she suggests the use of the linear mixed model. We have been searching in the literature and we have not been able to find even one reference supporting the use of this model in EWAS. We would like to make it clear that the aim of our work is not to explore new methodology in EWAS, but to perform one using the most solid pipeline as possible in order to report robust results. Anyway, we imagine that reviewer 4 suggests the use of linear mixed models in order to control for batch effect, but indeed our bioinformatics pipeline already controlled for batch effect using ComBat, a method widely employed in the field of microarrays. In addition to ComBat to control for batch effects, we used robust linear regressions. This statistical method is widely used in the PACE consortium, where very similar pipelines to the one used here have been implemented to find associations between DNA methylation levels of different tissues and multiple traits:

<https://www.niehs.nih.gov/research/atniehs/labs/epi/pi/genetics/pace/publications/index.cfm>.

We want to highlight that these studies have been recently published in high impact journals including several of the same editorial group as Commun Biol.

In summary, the inconsistency in the criteria and the lack of references to support the reviewers' statements has brought us to a dead end, and thus to the feeling of arbitrary opinions and bad luck are the ones deciding about our work. Not to mention that this process has lasted 9 months and 3 external revisions. This is a considerable time for PhD students and tenure track scientists, who have been involved in this project for several years, to achieve as a result the largest collaborative effort on placental methylation to date. As a consequence, we want to ask you to please reconsider your decision. Thank you.

Sincerely,

Jose Ramon Bilbao, University of the Basque Country (UPV/EHU)
Marie-France Hivert, Harvard Medical School
Nora Fernandez-Jimenez, University of the Basque Country (UPV/EHU)
Mariona Bustamante, ISGLOBAL
Ariadna Cilleros-Portet, University of the Basque Country (UPV/EHU)

El 17 jun 2022, a las 23:39, commsbio@nature.com escribió:

Dear Dr Bilbao,

Your manuscript entitled "Pre-pregnancy maternal body mass index and placental DNA methylation: a meta-analysis of 2,631 mother-child pairs" has now been seen by 2 referees, whose comments are appended below. We consulted with a new reviewer, who was asked to comments on the concerns of reviewer 1.

Unfortunately, in light of their advice I regret to inform you that we cannot publish your manuscript in Communications Biology. We understand that this decision is very disappointing to you but we are not able to take the paper forward given that the concerns regarding the statistics have not been addressed.

We feel that these reservations are sufficiently important as to preclude publication of this study in Communications Biology.

I am sorry that we cannot be more positive on this occasion and thank you for the opportunity to consider your work. We hope that you will find our referees' comments helpful when preparing your paper for submission elsewhere.

Best regards,

Christina Karlsson Rosenthal, PhD
Chief Editor
Communications Biology
orcid.org/0000-0002-3982-1784

Reviewers' comments:

Reviewer #3 (Remarks to the Author):

In this study, the authors investigate if associations between pre-pregnancy BMI (ppBMI) and various birth defects could be mediated by DNA methylation. Towards this, the authors make use of placenta methylation data from 2600 mother -child pairs from the PACE consortium.

While I cannot comment on the statistical analysis of this study, I find the presented results and conclusions are very

limited. The authors find 27 CpGs to be significantly associated with ppBMI. The reported differences in methylation 0.5-2% per 10 BMI units and I wonder how much such minimal changes in methylation can have any influence on downstream processes.

The follow-up analysis to interpret associations of these CpGs with nearby genes and regulatory elements and the drawn conclusions from this data are not very convincing, given the small amount of CpGs used.

Furthermore, the authors state in the abstract: "Our findings suggest that placental DNAm could be one of the mechanisms by which maternal obesity is associated with metabolic health outcomes in newborns and children". I suggest to rephrase to "we find correlations between placental DNAm, maternal obesity is associated with metabolic health outcomes in newborns and children" since the original sentence is misleading and no experiments are provided to validate this claim.

Reviewer #4 (Remarks to the Author):

The authors present a beautifully written manuscript and response to the previous round of review. As the research question was on maternal pre-pregnancy body mass index, I was very heartened that the authors performed methylation analysis in placental tissue, rather than blood. The authors further compared results with prior findings obtained from cord blood. In summary, the authors identified 27 CpGs that were associated with maternal ppBMI at experiment-wide significance. The gene and functional annotations provide helpful biological insights into the determinants of maternal ppBMI.

With regards to the statistical treatment of the data, i have to humbly agree with the points brought up by Referee #1. I particularly appreciate this reference that was raised by the authors in their response: "Robust Regression Methods: Achieving Small Standard Errors When There Is Heteroscedasticity", Rand R. Wilcox & H. J. Keselman". However, in a genome-wide search, certain additional safeguards should also be considered in the light of the very high genomic inflation (of 1.23 and 1.22) in the meta-analysis. Such a high residual inflation is normally unacceptably high in genome-wide studies (see a particularly vigorous discussion here: Sawcer S et al., Genetic risk and a primary role for cell-mediated immune mechanisms in multiple sclerosis. Nature. 2011; 476(7359):214-9). In such instances within the context of genome-wide genetic screens, where a very large number of markers are tested, other alternate methods of analysis could be considered and explored. One of these methods is the linear mixed model (described in detail in Sawcer et al., 2011), and i pray the authors could consider it.

Although we cannot publish your paper, it may be appropriate for another journal in the Nature Portfolio. If you wish to explore the journals and transfer your manuscript please use our manuscript transfer portal. You will not have to re-supply manuscript metadata and files, but please note that this link can only be used once and remains active until used. For more information, please see our manuscript transfer FAQ page.

Note that any decision to opt in to In Review at the original journal is not sent to the receiving journal on transfer. You can opt in to In Review at receiving journals that support this service by choosing to modify your manuscript on transfer. In Review is available for primary research manuscript types only.

This email has been sent through the Springer Nature Tracking System NY-610A-NPG&MTS

Confidentiality Statement:

This e-mail is confidential and subject to copyright. Any unauthorised use or disclosure of its contents is prohibited. If you have received this email in error please notify our Manuscript Tracking System Helpdesk team at <http://platformsupport.nature.com> .

Details of the confidentiality and pre-publicity policy may be found here
<http://www.nature.com/authors/policies/confidentiality.html>

Privacy Policy | Update Profile

REVIEWERS' COMMENTS:

Reviewer #5 (Remarks to the Author):

Thank you for the opportunity to review the manuscript entitled: Pre-pregnancy maternal BMI and placental DNA methylation: a meta-analysis of 2,631 mother-child pairs. The report presents the findings of the largest meta-analysis of placental DNA methylation in relation to maternal BMI in predominantly European population. The report presents 27 CpGs identified, compares these with findings of DNA methylation in the cord blood, and also presents findings from downstream analyses. The large sample size and adjustment to several important covariates make the study robust although there are some issues that need to be clarified to improve the scientific value of this work.

Major issues:

1. The pre-pregnancy maternal BMI is the main exposure variable. The quality of assessment is doubtful as it is based on self-report. Also, for unavailable pre-pregnancy BMI, values in the first trimester were used. This is an important limitation which needs to be addressed in the discussion. It will be helpful to present comparison of BMI between those with reports from before pregnancy and those from the first trimester to rule out the possibility of systematic selection. The authors may also consider the timing of measurement of BMI as co-variate.
2. Population structure may influence the results and adjusting the models to genotype principal components provides robust results even when the samples are from similar ancestry. If genotypes are unavailable, the limitations including potential generalizability of the findings for non-European populations where obesity is a much more common issue needs to be discussed.

Minor issues

1. The authors made comparisons between cord blood DNA methylation and placental DNA methylation in the context of similar environmental exposure - maternal BMI. The authors suggest that the DNA methylation changes in the placenta as potential mechanisms for the link between maternal obesity and long-term offspring health outcomes. While this is possible, the authors fail to mention that the DNA methylation changes in placenta could simply be markers of exposure to maternal high BMI.
2. There are typographical and grammatical errors that require correction e.g. line 237: 'performed'

Pre-pregnancy maternal body mass index and placental DNA methylation: a meta-analysis of 2,631 mother-child pairs

Response to Reviewer #5

Thank you for the opportunity to review the manuscript entitled: Pre-pregnancy maternal BMI and placental DNA methylation: a meta-analysis of 2,631 mother-child pairs. The report presents the findings of the largest meta-analysis of placental DNA methylation in relation to maternal BMI in predominantly European population. The report presents 27 CpGs identified, compares these with findings of DNA methylation in the cord blood, and also presents findings from downstream analyses. The large sample size and adjustment to several important covariates make the study robust although there are some issues that need to be clarified to improve the scientific value of this work.

Response: We thank the reviewer's words and advice.

Major issues:

1. The pre-pregnancy maternal BMI is the main exposure variable. The quality of assessment is doubtful as it is based on self-report. Also, for unavailable pre-pregnancy BMI, values in the first trimester were used. This is an important limitation which needs to be addressed in the discussion. It will be helpful to present comparison of BMI between those with reports from before pregnancy and those from the first trimester to rule out the possibility of systematic selection. The authors may also consider the timing of measurement of BMI as co-variate.

Response: We agree with the reviewer that this is a limitation of our study. Anyway, if there is any effect between self-reported and measured BMI, this effect will be by cohort (as most individuals in each cohort were assessed in the same way), and this effect would have arisen in the leave-one-out meta-analysis. Additionally, we performed a comparison in the INMA cohort and found out that self-reported ppBMI was highly correlated with measured BMI at 12 weeks of gestation ($r = 0.96$; $P < 0.0001$). We have disclosed this limitation in the article by adding:

“On the one hand, we are aware that self-reported ppBMI may not be the most accurate measurement for our variable of interest, as may also be the case of the measurement of BMI at the end of the first trimester. However, self-reporting of ppBMI has been shown to be reliable and highly correlated to measured BMI at 12 weeks of gestation ($r=0.96$; $P < 0.0001$).”

Casas M, et al. Maternal pre-pregnancy overweight and obesity, and child neuropsychological development: two Southern European birth cohort studies. Int J Epidemiol. 2013 Apr;42(2):506-17. (Added to the bibliography of the manuscript)

2. Population structure may influence the results and adjusting the models to genotype principal components provides robust results even when the samples are from similar ancestry. If genotypes are unavailable, the limitations including potential generalizability of the findings for non-European populations where obesity is a much more common issue needs to be discussed.

Response: It is very true, as pointed out by the reviewer, that this is another limitation of our work. In fact, although as far as we have been able to assess, all (or the great majority of) participants are of European descent, adding several Principal Components of the genotype would have been safer, but several cohorts did not have genotype data available and we wanted to keep the models stable across cohorts. Additionally, the relative homogeneity of all our datasets, in terms of ancestry, makes it hard to

generalize our findings to other non-European populations. We have added this consideration to the discussion:

“Second, the unavailability of genotype data in some of the participating cohorts did not allow to add genotype principal components to our models and there might be residual confounding by population structure that we did not account for. Third, most of the cohorts were composed by a majority of individuals from European descent, which limits generalizability of our findings to other populations.”

Minor issues:

1. The authors made comparisons between cord blood DNA methylation and placental DNA methylation in the context of similar environmental exposure - maternal BMI. The authors suggest that the DNA methylation changes in the placenta as potential mechanisms for the link between maternal obesity and long-term offspring health outcomes. While this is possible, the authors fail to mention that the DNA methylation changes in placenta could simply be markers of exposure to maternal high BMI.

Response: *The reviewer is again right, and thus, we have added this sentence in the conclusion:*

“...we cannot rule out that the changes observed could be markers of exposure to high ppBMI and therefore, our findings will need to be supplemented by functional studies...”

2. There are typographical and grammatical errors that require correction e.g. line 237: 'preformed'

Response: *We apologize and have thoroughly reviewed the manuscript and believe to have corrected typos and errors accordingly.*